# Semantic Information-enhanced Loop Closure Detection for Simultaneous Localization and Mapping

1st Yushan Huang
*Basic research center*
*University of Electronic Science and Technology of China*
Cheng Du, China
202222060330@std.uestc.edu.cn

2nd Zhifeng Wang
*School of automation Engineering*
*University of Electronic Science and Technology of China*
Cheng Du, China
202221060533@std.uestc.edu.cn

3rd Yaoyu Ding
*School of automation Engineering*
*University of Electronic Science and Technology of China*
Cheng Du, China
yding@uestc.edu.cn

4th Lu Yang
*Basic research center*
*University of Electronic Science and Technology of China*
Cheng Du, China
yanglu@uestc.edu.cn

5th Jinliang Shao
*Basic research center*
*University of Electronic Science*
*and Technology of China*
Cheng Du, China
jinliangshao@126.com

*Abstract*—In visual navigation, Simultaneous Localization and Mapping (SLAM) faces the pivotal challenge of loop closure detection, which is vital for refining position estimates and map construction. Current approaches relying on deep learning-based global descriptors struggle with robustness and interpretability. To overcome these limitations, we propose a novel method that integrates partial semantic segmentation with traditional location recognition networks through a weighted fusion mechanism. By harnessing the synergy of semantic and spatial information, our approach provides deeper insights into image content and spatial relationships. The carefully crafted weighting scheme enables a more comprehensive assessment of image similarity, considering both the "what" and "where" of image features. Experimental evaluations conducted on the Pittsburgh 250k dataset, comprising an extensive collection of 250,000 images, consistently showcase the effectiveness of our fusion strategy. Across all three tested backbone networks, we observe a notable improvement of over 3% in recall rate for loop closure detection. Notably, when employing MobileNet as the backbone, the enhancement is even more pronounced, surpassing 5% with an optimal configuration featuring a semantic vector weight of 0.94 and a location network weight of 0.06. This significant achievement not only underscores the robustness and accuracy gains achievable through our approach within SLAM systems but also highlights its potential as a versatile strategy for semantic-spatial integration, with promising applications in various computer vision tasks that require advanced spatial-semantic comprehension.

*Keywords—Semantic Information, Loop Closure Detection, VLAD Encoding, Global Semantic Descriptors, Weighted Fusion, Visual SLAM*

## I. INTRODUCTION

Visual Simultaneous Localization and Mapping (SLAM) technology, as one of the core technologies in the field of visual navigation [1,25], currently holds a pivotal position in robotic applications. The mainstream visual SLAM solutions rely on continuous image sequences captured by cameras, utilizing geometric principles to estimate the motion changes between adjacent images and optimizing the backend of the system to enhance the accuracy of localization and mapping [2], [24].

Within this process, Loop Closure Detection [3,26] plays a crucial role as a key support technology for backend optimization. Loop Closure Detection aims to identify whether the camera has revisited a previous location, providing vital global consistency constraints for the backend optimization process. Ensuring the accuracy of loop closure detection algorithms hinges on effectively reducing false positives (incorrectly identified loops) and false negatives (missed true loops), with achieving accurate and efficient detection being the core challenge in algorithm design.

Currently, loop closure detection algorithms are widely researched and applied in the visual SLAM field, with their implementation strategies primarily categorized into three types: (1) Map-to-Map methods, represented by the "segmented" map concept proposed by Clemente and Davison in 2007 [4], which significantly enhances stability in complex dynamic environments. However, relying on sparse map construction, these methods suffer from limited spatial feature information, potentially leading to inaccurate loop closure results due to insufficient information. (2) Image-to-Map methods, where Williams et al. [5] introduced a method to determine loop relationships by matching current image frames with map features in designing a relocalization module. Although intuitive and effective, this approach wastes computational resources as it requires training classifiers with all loop information for each loop detection. (3) Image-to-Image methods, currently the most popular loop closure detection strategy, focus on assessing image similarity using feature

descriptors, which can be further divided into local [6] and global descriptors [7]. Global descriptors like PCA (Principal Component Analysis) [8] describe entire images with a single global descriptor but are susceptible to environmental changes. Local descriptor methods, especially when combined with the Bag-of-Words (BoW) technique, compare visual words in images to filter loop candidates and verify loop authenticity with global descriptors. The DBoW (Direct Binary Bag-of-Words) algorithm by Galvez-Lopez and Tardos [9] further enhances computational efficiency by vectorizing images through a vocabulary tree and utilizing forward and reverse index structures, effectively addressing the issue of traditional algorithms' increasing time consumption with the number of images. Subsequently, NetVLAD [10], introduced in 2016 as a scene recognition algorithm, revolutionized traditional VLAD algorithms. While the original VLAD algorithm relies on SIFT or similar algorithms as its foundation, encoding the features generated by these algorithms into a concise feature vector, NetVLAD ingeniously integrates this process with Convolutional Neural Networks (CNNs), leveraging CNNs as powerful feature extraction tools to construct an end-to-end trainable system. In short, by introducing deep learning techniques, particularly CNNs, NetVLAD optimizes feature extraction and encoding, enabling more efficient and accurate scene recognition.

In summary, while progress has been made in the field of loop closure detection, even advanced algorithms like NetVLAD face numerous challenges and issues. Firstly, the accuracy of loop closure detection in complex and dynamic scenarios remains insufficient, directly impacting the performance of the entire SLAM system. Specifically, a correctly detected true positive loop significantly improves the precision of visual odometry and reduces accumulated errors, whereas a false positive loop (incorrectly identified) may guide the backend optimization module towards erroneous convergence, negatively affecting the stability and accuracy of the entire SLAM system. Secondly, for deep learning algorithms like NetVLAD, while they can automatically learn complex feature representations from vast amounts of data, their decision-making processes often lack transparency, making them difficult for humans to directly comprehend. This leads to difficulties in tracing the root causes of false positive loops and effectively adjusting algorithm parameters or structures to avoid similar errors.

Addressing the challenges of insufficient accuracy and interpretability in loop closure detection, this paper proposes an innovative loop closure detection algorithm based on an in-depth study of existing algorithms. This algorithm not only inherits the advantages of deep learning in feature extraction, effectively capturing global scene features through global descriptors, but also innovatively introduces an image semantic verification module to enhance the algorithm's robustness and interpretability. This module refines loop candidates obtained through global descriptor matching by comparing the consistency and similarity of semantic information between candidate loop frames and reference frames. To achieve this, a novel method combining deep semantic networks and VLAD encoding technology is proposed, aiming to generate robust and discriminative global semantic descriptors. Initially, rich semantic features are extracted from images using pre-trained deep semantic networks. Subsequently, these local semantic features are aggregated into a compact global semantic descriptor through VLAD encoding technology.

The major contributions are given as follows:

- Generation of Global Semantic Descriptors: This method commences by extracting rich semantic features from images using a pre-trained deep semantic network. Subsequently, these features are aggregated into a compact global semantic descriptor through a custom-trained VLAD encoding process. This custom VLAD network is specifically trained to optimize the representation of the semantic features for the task at hand.

- Novel fusion:Our approach integrates the visual descriptors, typically derived from traditional place recognition networks, with the global semantic descriptors generated by the custom-trained VLAD network through a weighted fusion approach.The weights for this fusion process are also optimized through training, ensuring a more precise assessment of inter-frame similarity.

- Experimental Validation: Extensive experiments conducted on three distinct backbone networks reveal that the integration of the custom-trained VLAD model and optimized fusion strategy significantly enhances the performance of the recognition system. Notably, the MobileNet-based recognition network achieves a remarkable 5% boost in recall rate, while the other two backbone networks also demonstrate improvements of over 3%, underscoring the algorithm's efficiency and robust adaptability to diverse network architectures.

## II. RELATED WORK

With the rapid advancements in computer vision and deep learning, a surge of research based on deep learning has emerged in the field of computer vision. An increasing number of researchers are leveraging deep learning algorithms to extract semantic information embedded in the environment, facilitating high-level understanding of scenes and integrating the acquired semantic information with SLAM (Simultaneous Localization and Mapping) technology to build semantic SLAM systems.

The first step in semantic VSLAM is to extract semantic information from images captured by cameras. By classifying image information, semantic information derived from image content can be obtained [11]. In the early days, the only available method for semantic information extraction was object detection, which relied on interpretable machine learning classifiers such as decision trees and support vector machines for classifying and extracting objects. However, with technological evolution, modern semantic VSLAM systems have increasingly adopted deep learning techniques to construct semantic extraction modules like object detection and semantic segmentation for extracting semantic information from images [23].

Ren et al. proposed Faster R-CNN, unifying the fundamental steps of object detection into a single deep network framework, significantly enhancing training and testing efficiency [12]. Mask R-CNN, introduced by He et al., is a paradigmatic application of the Faster R-CNN concept in instance segmentation. Its core idea is to augment the target classification and regression branches with a semantic segmentation branch to predict regions of interest, utilizing Fully Convolutional Networks (FCN) to predict the category of each pixel [13]. Wang et al. presented RDS Net, which incorporates three modules: a mask refinement and target localization module, a mask pruning module, and a target frame-assisted instance mask relationship module. Its two-stream network design substantially addresses the issue of low resolution in instance masks [14]. Cai et al. proposed Cascade R-CNN, extending the cascade architecture to image segmentation tasks by integrating a segmentation branch at each cascade stage [15]. Hurtik et al. improved upon YOLOv3 by introducing poly-YOLO, which addresses the issues of extensive label overwriting and inefficient anchor distribution in YOLOv3, enhancing accuracy while reducing parameters [16]. Instance segmentation tasks have also found widespread applications in scenarios such as remote sensing images and face detection.

In the research field of utilizing semantic information to assist Simultaneous Localization and Mapping (SLAM), several pioneering works have demonstrated how semantic information significantly enhances the performance and intelligence of SLAM systems. Specifically, Wen et al. [17] proposed a semantic topological map framework based on a binocular visual-inertial SLAM system, adopting a hybrid 3D point cloud semantic topological map construction framework for autonomous navigation and loop closure detection. Han and Xi [18] introduced a semantic SLAM method for dynamic environments, utilizing optical flow to identify and exclude dynamic points, and treating feature points located on dynamic objects as dynamic points for exclusion. Furthermore, semantic information was used to generate point cloud maps and semantic octree maps. Cheng et al. [19] combined deep learning with visual SLAM, constructing a semantic map of the environment while simultaneously employing an optical flow-based approach to handle dynamic objects, enabling the system to operate in dynamic environments.

Currently, although research in the semantic SLAM domain has made initial progress, existing works primarily focus on integrating semantics into localization and mapping. In contrast, research on effectively applying semantic information to loop closure detection remains relatively scarce. Loop closure detection, as a crucial component of SLAM systems, is essential for eliminating accumulated errors and constructing globally consistent maps. If semantic information could be more fully exploited, by recognizing objects, scenes, and their relationships in the environment, it would undoubtedly further improve the accuracy and robustness of loop closure detection. As a result, the field of semantic SLAM research offers ample opportunities for advancement and refinement [20-22].

## III. SEMANTIC INFORMATION-ENHANCED LOOP CLOSURE DETECTION

This section centers on the localization aspect of the Semantic SLAM system, specifically enhancing the loopback detection mechanism beyond conventional visual SLAM. Rather than detailing the pose estimation methods of traditional SLAM, we integrate semantic information to introduce a novel loop detection approach tailored for our Semantic SLAM system. As illustrated in Fig. 1, the methodology is concisely outlined, showcasing how semantic cues augment the loopback detection process, enhancing its robustness and accuracy.

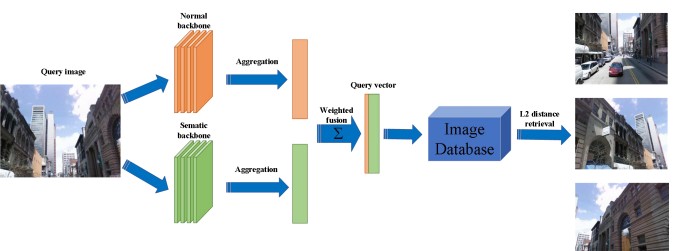

Fig. 1. An overview of our proposed semantic information-enhanced loop closure detection method

### A. Problem Formulation and Training Objective

The framework outlined incorporates two primary phases: global retrieval and semantic consistency constraint. The aim of loop closure detection is to devise an embedding space, given a collection of query images $\{ I_q \}$ and reference images $\{ I_r \}$, where each query image $I_q$ is proximally positioned to its corresponding positive reference image $I_r$. During training, reference images that share the same location as the query images are classified as positive samples, adhering to a standard threshold of 10 meters. In line with prior research [27], we designate the reference image nearest to each query in the embedding space as the definitive positive sample. Conversely, reference images positioned more than 25 meters away are designated as negative samples.

To bolster training efficiency and the model's ability to distinguish between samples, we implement partial negative mining [27], a strategy that selects the most challenging negative samples from a randomly selected subset. The global embedding representations of the queries, positive samples, and negative samples are labeled as $E_q$, $E_p$, and $E_n$, respectively. To refine the model, we optimize the global retrieval loss utilizing a margin triplet loss function.

$$L_g = \max(m + \|E_q - E_p\|^2 - \|E_q - E_n\|^2, 0). \quad (1)$$

In this context, $\|*\|^2$ stands for the square of the L2 norm, and m denotes the boundary margin.

### B. Global Retrieval Module

The Global Retrieval Module (GRM) serves as a pivotal component in our proposed system, tasked with extracting and integrating global features from images or objects to facilitate efficient classification, recognition, and matching tasks. This module leverages the robust feature extraction capabilities of

deep learning models, coupled with advanced feature aggregation techniques, to generate highly representative and robust global descriptors.

- Deep Feature Extraction:Firstly, a pre-trained deep neural network (such as convolutional neural network CNN) is used as the feature extractor to process the input image layer by layer, and gradually abstract the feature representation of the image from the bottom layer to the top layer.

- Feature Aggregation and Encoding:After extracting the local features, they are subsequently fed into the feature aggregation and encoding stage, where we employ VLAD (Vector of Locally Aggregated Descriptors) as the primary technique. VLAD is a powerful method that effectively converts local features into a fixed-length global descriptor while preserving the image's overall characteristics and critical information.

- Global Retrieval:The generated global descriptors are stored in a database equipped with a corresponding indexing mechanism. Upon receiving a query, the system computes the global descriptor of the query image and utilizes efficient similarity metrics (such as cosine similarity or Euclidean distance) to alculate the global descriptor score.

## C. Semantic consistency constraint

To overcome the limitations of traditional global descriptors in capturing robust and semantically meaningful representations of images, we propose a Semantic Consistency Constraint that integrates deep semantic information with VLAD encoding. This constraint serves as a refinement step after initial global descriptor matching, ensuring that only semantically consistent and similar frames are considered as valid loop closure candidates.

- Semantic Segmentation:We utilize a pre-trained semantic segmentation model, such as Deeplab, to extract rich semantic features from the input images. This model is capable of identifying different regions within an image and assigning them to specific semantic categories, providing a detailed understanding of the scene content.

- Feature Selection and Processing:From the output of the semantic segmentation model, we select the most representative feature maps that contain rich information about key semantic categories. These feature maps undergo further processing, including pooling and normalization, to reduce their dimensionality while preserving the most important information for downstream tasks.

- VLAD Encoding Layer for Semantic Similarity:To harness the rich semantic information extracted from images and encode it into a compact yet discriminative global representation, we introduce a customized VLAD (Vector of Locally Aggregated Descriptors) encoding layer tailored specifically for our framework. This layer serves as a pivotal step in our proposed method, transforming processed semantic features into a semantic global descriptor that enables precise semantic similarity computations.

- End-to-End Training:Instead of training the semantic segmentation model and VLAD layer separately, we adopt an end-to-end training approach. This allows us to optimize both components jointly, ensuring that the generated global semantic descriptors are tailored specifically for the loop closure task. During training, we minimize a loss function that captures both the accuracy of the semantic segmentation and the discriminative power of the generated descriptors.

## D. Weighted Fusion

Our method is specifically designed to enhance the recall rate of loop closure detection, which is crucial for ensuring that all actual loop closures are identified without missing any. Rather than solely focusing on accuracy, we emphasize the importance of minimizing false negatives (missed loop closures) through a comprehensive approach.

In this step, we integrate visual and semantic information in a novel way.A global visual descriptor is extracted from the reference frame and compared to candidate frames using a suitable metric, denoted by the symbol $G_{A,B}$ . Additionally, we analyze semantic features to generate a semantic similarity score $S_{A,B}$ , which captures deeper understanding of the scene context and further aids in the detection process. By combining the visual similarity score $G_{A,B}$ and the semantic similarity score $S_{A,B}$ , through weighted fusion, we obtain a composite similarity score $C_{A,B}$ . This process integrates both visual and semantic information to provide a holistic assessment of the similarity between frames.

$$C_{A,B} = (1 - \alpha)G_{A,B} + \alpha S_{A,B} \tag{2}$$

Here, $\alpha$ serves as a tuning knob, allowing the system designer to emphasize either the visual or semantic component as needed. By strategically adjusting $\alpha$, the system can tailor its loop closure detection to specific environments or requirements. For instance, if semantic information is deemed more crucial in a given application, $\alpha$ can be set to a higher value (close to 1).

The advantage of this approach lies in its ability to harness the complementary strengths of visual and semantic information. While visual similarity helps identify structural or textual similarities, semantic similarity adds a layer of robustness by capturing meaning beyond mere visual appearances. This is due to the fact that images captured from adjacent locations but with different viewpoints retain semantic similarity. Even when the camera is moving at a relatively fast speed, the semantic information described in adjacent images remains largely consistent. By harnessing the power of semantics, our method achieves a heightened sensitivity to loop closures, especially in complex and dynamic environments, significantly boosting the recall rate and ensuring that genuine loop closures are rarely overlooked.

## IV. Experiments and Results

### A. Improving the Experimental Environment for loop detection

To validate the effectiveness of our proposed method for enhancing loop detection in Simultaneous Localization and Mapping (SLAM) systems, we utilized the comprehensive and challenging Pittsburgh 250k dataset. This dataset, sourced from Google Street View, comprises a vast collection of 250,000 panoramic images that capture diverse street-level views of Pittsburgh, Pennsylvania, USA. These images not only offer rich visual features but also exhibit significant variations in lighting conditions, seasonal changes, and viewpoint angles, making it an ideal testbed for evaluating the robustness and accuracy of loop detection algorithms.

Specifically, we leveraged the 24,000 query images provided in the dataset, which were captured at different times from the database images, to simulate real-world conditions where loop candidates may appear significantly dissimilar due to temporal gaps. By employing this large-scale and realistic dataset, we aimed to create a more stringent experimental environment that could thoroughly assess the performance of our weighted semantic-visual fusion approach.

Moreover, the Pittsburgh 250k dataset allows for a comprehensive evaluation of the recall rate, a crucial metric in loop detection, as it enables the identification of true positive loops amidst a vast pool of potential candidates. By analyzing the performance of our method on this dataset, we were able to demonstrate its capability to significantly improve the recall rate compared to traditional approaches, thereby enhancing the overall robustness and reliability of SLAM systems.

### B. Evaluation Metrics

Two evaluation metrics, namely Precision and Recall, can be utilized to quantify the accuracy of loopback detection, based on its categorization. These metrics are determined through the following calculations:

$$\begin{cases} Precision = \frac{TP}{TP+FP} \\ Recall = \frac{TP}{TP+FN} \end{cases} \quad (2)$$

Precision quantifies the fraction of correctly identified loop closures among all the detected loops, indicating the likelihood that a detected loop is genuinely a true loop. Recall, on the other hand, measures the proportion of actual true loop closures that have been successfully detected, reflecting the capability of the system to find all existing true loops.

### C. Results and discussion

In this study, we delved into the influence of semantic cues on the recall rate of loop closure detection within the context of place recognition systems. To achieve this, we employed several prevalent deep learning backbones, namely ResNet, VGG16, and MobileNet, as the foundations for our experimental frameworks. These backbones were selected due to their varying complexity and capabilities in extracting robust visual features, enabling us to evaluate their potential contribution to enhancing loop closure detection performance.

Our core approach involved leveraging partial semantic segmentation outcomes and fusing them with the outputs of traditional location recognition networks through a weighting mechanism. Specifically, we use the Faiss library to efficiently perform similarity search on large-scale vector data. With L2 distance as the similarity metric, the most similar vector sets are successfully retrieved from massive data for given queries. This integration strategy aimed to harness the complementary strengths of both semantic and spatial information, ultimately boosting the system's ability to accurately detect loop closures.

In this study, we have set different a values for distinct backbone architectures to optimize their performance. Specifically, we adopted an α value of 0.938 for the ResNet model, 0.920 for the VGG16 model, and 0.940 for MobileNet. The results of our experiments, presented in Tab. 1, demonstrate a marked improvement in the recall rate including @1, @5, and @10, when partial semantic results are incorporated into the decision-making process, as compared to solely relying on either traditional place recognition networks or purely semantic networks.

TABLE I. Results

| Recall(@1,5,10) | Distinct Backbone Architectures | | |
|---|---|---|---|
| | ResNet50 | VGG 16 | MobileNet |
| Semantic | 0.4278/0.6401/0.7196 | 0.3971/0.5394/0.6512 | 0.3833/0.5670/0.6688 |
| Normal | 0.6165/0.8297/0.8658 | 0.7917/0.9024/0.9319 | 0.4949/0.7359/0.8214 |
| Fusion (ours) | 0.6852/0.8530/0.9007 | 0.8173/0.9372/0.9653 | 0.6067/0.8025/0.8743 |

### D. Qualitative evaluation

In this section, we present a qualitative assessment of our proposed integration strategy that leverages partial semantic segmentation outcomes and fuses them with traditional location recognition networks. This evaluation aims to provide insights into how the fusion of semantic and spatial information enhances the system's ability to accurately detect loop closures, beyond the quantitative improvements demonstrated in the previous section.

*1) Fusion results analysis:* First, we analyze the results using the provided method. Fig. 2 showcases the results of combining semantic and spatial information in detecting loop closures. Specifically, we observe that in complex environments with similar visual appearances but distinct semantic contexts, the fusion method is able to distinguish between these scenarios more effectively than either the purely semantic or purely spatial approach alone. The semantic information helps in identifying unique objects or features that are indicative of a specific location, while the spatial information provides context about the relative positions and orientations of these features.

Query  Retrieval @1  Retrieval @2  Retrieval @3  Retrieval @4

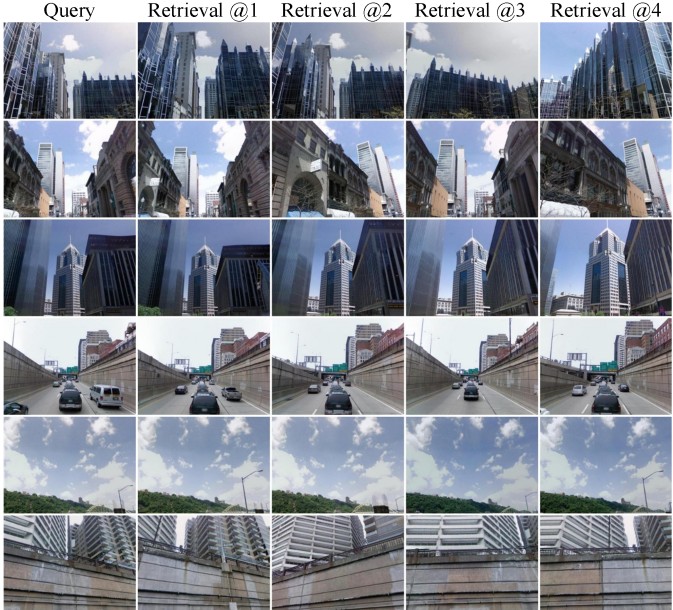

Fig. 2. Our fusion results

*2) Ablation studies:* To further illustrate the complementary nature of semantic and spatial information, we present the results of two ablation studies: one using only semantic information and the other using only spatial information.

*a) Semantic-only approach:* In the purely semantic approach, as Fig. 3 shows, we observe that while the system is capable of recognizing different objects and features within the environment, it encounters difficulties in scenarios where the semantic content is similar but the spatial layouts differ. For instance, two images depicting roads and houses with similar semantic information but distinct spatial details may be mistakenly classified as the same location due to the absence of spatial context. This underscores the limitation of relying solely on semantic information for loop closure detection, highlighting the need for additional cues to accurately distinguish between similar yet spatially distinct environments.

Query  Retrieval @1  Retrieval @2  Retrieval @3  Retrieval @4

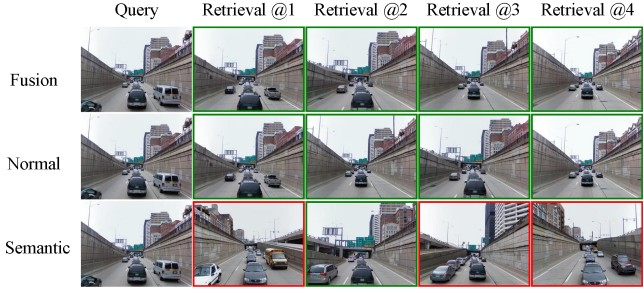

Fig. 3. Semantic-only results

*b) Spatial-only approach:* On the other hand, the spatial-only approach, while adept at capturing the macro-level layout and geometry of the environment, falters in fine-grained distinctions where spatial layouts are nearly identical

but semantic nuances diverge significantly. For instance, in our experiments(Fig. 4), we encountered cases where two images—both showcasing a similar composition with an upper half dominated by the sky, a verdant green landscape in the middle, and a road network in the lower portion—were erroneously matched as loop closures due to their striking similarity in spatial configuration. Despite these visual similarities, the presence of key semantic elements like bridges, pedestrian walkways, or specific lane configurations varied significantly between the images. The inability of the spatial-only approach to discern these critical semantic differences undermined its accuracy in distinguishing between visually similar yet semantically distinct locations. This highlights the pivotal role that semantic information plays in augmenting spatial data. By incorporating semantic understanding into the system, it becomes possible to recognize the subtle yet important differences that truly distinguish one location from another. This, in turn, enables more accurate loop closure detection, avoiding erroneous matches and improving the overall performance of the approach. Therefore, combining spatial and semantic information is crucial for developing robust and reliable systems that can accurately navigate and map complex environments.

Query  Retrieval @1  Retrieval @2  Retrieval @3  Retrieval @4

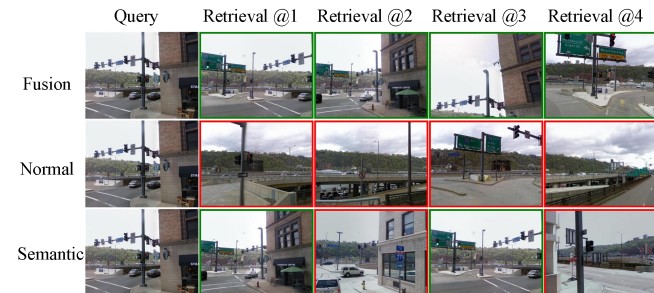

Fig. 4. Normal(spatial-only) results

## V. CONCLUSION

This study has investigated the impact of incorporating semantic information, through the utilization of pre-trained semantic segmentation networks (specifically, Deeplabv3 and UNet), on the recall rate of loop closure detection within SLAM systems. By leveraging three distinct backbone architectures—ResNet50, VGG16, and MobileNet—we conducted a comprehensive evaluation to assess the effectiveness of semantic cues in enhancing the performance of loop closure detection.

Our findings reveal a pivotal observation: the fusion of partial semantic results with the outputs of traditional location recognition networks, through a weighted approach, leads to a substantial improvement in the recall rate of loop closure detection. The results underscore the complementary nature of semantic and spatial information in loop closure detection, demonstrating that the integration of these modalities can significantly enhance the precision and robustness of SLAM systems. This work not only challenges the conventional wisdom of limited semantic utilization in traditional SLAM

frameworks but also presents a promising direction for future optimizations and upgrades, emphasizing the potential of multi-modal fusion strategies to advance the state-of-the-art in loop closure detection and, consequently, the overall performance of SLAM systems.

By highlighting the positive influence of semantic information on recall rate, our study contributes to the growing body of research exploring the integration of semantic cues into SLAM pipelines, offering valuable insights for researchers and practitioners alike.

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
