# OpenReview forum: "Semantic Information-enhanced Loop Closure Detection for Simultaneous Localization and Mapping"
_IEEE.org/ICIST/2024/Conference — IEEE ICIST 2024 Conference Submission_

### Official Review · Reviewer_3fJC · 2024-08-22
**Promising method**

**Rating:** 7
**Confidence:** 3

**Review:**

The paper introduces a promising method for improving loop closure detection in SLAM systems by integrating semantic segmentation with traditional location recognition networks through a weighted fusion mechanism. This topic is interesting, the following comments need to further consider: a. Provide more detail on how the weighted fusion mechanism operates. This could include a brief explanation of how the weights are assigned or how the integration of semantic and spatial data is technically achieved. b. Incorporate a brief mention of how the proposed method compares with existing approaches. Highlighting specific improvements in accuracy, robustness, or computational efficiency would provide a clearer understanding of the method's contributions. c. Consider simplifying some of the technical terms or providing brief explanations. This would make the abstract more accessible without losing its relevance to the intended audience. d. The format of the references should be unified.

---

### Official Review · Reviewer_9jiu · 2024-08-22
**accept**

**Rating:** 7
**Confidence:** 4

**Review:**

This paper introduces a method that integrates partial semantic segmentation outcomes with traditional location recognition networks through a novel weighted fusion mechanism.
1). In abstract, about the proposed method, the statement is unclear. Authors need to rewrite abstract and to focus on the proposed method and to stress both the specific application and the generic aspects of the work.
2). The innovation of this article is written in a rather cumbersome manner and needs to be simplified.

---

### Official Review · Reviewer_hEFu · 2024-08-24
**This article is very interesting and a good one.**

**Rating:** 7
**Confidence:** 5

**Review:**

This paper introduces a method that integrates partial semantic segmentation outcomes with traditional location recognition networks through a novel weighted fusion mechanism. The obtained result is valuable and can be accepted if the following problems can be clarified.
1. What's the difference between the 'Introduction' and the 'Related Works' ?
2. Please further explain how the loop closure detection algorithm proposed in this paper enhance the robustness and interpretability.
3. The format of references needs to be uniform.

---

### Decision · Program_Chairs · 2024-09-08

Accept (Oral)